# Method for Using IMU-Based Experimental Motion Data in BVH Format for Musculoskeletal Simulations via OpenSim

**DOI:** 10.3390/s23125423

**Published:** 2023-06-08

**Authors:** Iris Wechsler, Alexander Wolf, Sophie Fleischmann, Julian Waibel, Carla Molz, David Scherb, Julian Shanbhag, Michael Franz, Sandro Wartzack, Jörg Miehling

**Affiliations:** 1Engineering Design, Department of Mechanical Engineering, Friedrich-Alexander-Universität Erlangen-Nürnberg, 91058 Erlangen, Germany; wechsler@mfk.fau.de (I.W.);; 2Machine Learning and Data Analytics Lab, Department Artificial Intelligence in Biomedical Engineering (AIBE), Friedrich-Alexander-Universität Erlangen-Nürnberg, 91052 Erlangen, Germany

**Keywords:** biomechanics, musculoskeletal modelling and simulation, inertial sensors, motion tracking, data transfer

## Abstract

Biomechanical simulation allows for in silico estimations of biomechanical parameters such as muscle, joint and ligament forces. Experimental kinematic measurements are a prerequisite for musculoskeletal simulations using the inverse kinematics approach. Marker-based optical motion capture systems are frequently used to collect this motion data. As an alternative, IMU-based motion capture systems can be used. These systems allow flexible motion collection without nearly any restriction regarding the environment. However, one limitation with these systems is that there is no universal way to transfer IMU data from arbitrary full-body IMU measurement systems into musculoskeletal simulation software such as OpenSim. Thus, the objective of this study was to enable the transfer of collected motion data, stored as a BVH file, to OpenSim 4.4 to visualize and analyse the motion using musculoskeletal models. By using the concept of virtual markers, the motion saved in the BVH file is transferred to a musculoskeletal model. An experimental study with three participants was conducted to verify our method’s performance. Results show that the present method is capable of (1) transferring body dimensions saved in the BVH file to a generic musculoskeletal model and (2) correctly transferring the motion data saved in the BVH file to a musculoskeletal model in OpenSim 4.4.

## 1. Introduction

### 1.1. General Background

Biomechanical simulation enables in silico estimations of biomechanical parameters such as muscle, joint and ligament forces. Musculoskeletal models, which depict the human body as a multi-body system, are used to carry out the simulations. This makes the models an interesting tool for different scientific questions throughout many disciplines. Musculoskeletal simulations are utilized for medical investigations such as knee- or hip-implant simulations [1,2,3], for sport science tasks such as understanding body mechanisms in order to prevent sports injuries [4,5] or design tasks such as exoskeleton design [6,7,8], sports equipment design [9,10] or vehicle design [11]. One standard approach to computing biomechanical simulations is the inverse approach—inverse kinematics followed by inverse dynamics—which requires experimental kinematic measurements as input. By convention, optical motion capture systems are used to collect motion data. Hereby, markers are placed on specific locations on the body (mostly on anatomical landmarks). Multiple cameras are then needed to obtain three-dimensional marker trajectories using triangulation. The marker trajectories are based on the motion of the participant. For every time step of the measurement, the global position of each marker is saved in a resulting marker file. Although the marker-based approach is considered the gold standard for capturing motion, it also has some limitations. Since optical motion capture systems require specific camera systems, measurements need to be conducted in specifically equipped (gait) laboratories. This restrains the applicability of musculoskeletal simulations since not everybody (companies, private individuals, researchers, etc.) working with musculoskeletal human models has access to the required equipment/facilities, or the activities of interest cannot be measured in an optical motion capture setup, e.g., skiing [12]. In addition, marker occlusion—a state in which a marker is not visible to more than one camera because it is obscured by body parts or other markers—can occur. Thus, during the last decade, the popularity of wearable inertial measurement units (IMUs) has increased [13,14,15]. The advantage of a wearable IMU system is its flexible applicability with nearly no restrictions regarding the environment. Further, the marker placement process, which is quite time-consuming, is avoided. IMUs consist of a three-axis gyroscope and a three-axis accelerometer. Additionally, a three-axis magnetometer may be included. To store the IMU-based motion data, different data formats exist. A text-based file format such as txt or csv may be used to save either the unfiltered or filtered raw sensor data (acceleration, angular rate and optional magnetic field measurements) directly. Alternatively, the sensor data can be pre-processed by the motion capture system and be exported in a different file format, such as the BVH format, a universal motion capture format that provides information about both the underlying skeletal system and the captured motion simultaneously. Despite the advantages of IMU-based motion capture systems and the possibility of accessing raw or filtered sensor data directly, only a few approaches exist to drive musculoskeletal simulations with IMU data.

### 1.2. State of the Art

OpenSense, a toolkit implemented in the software OpenSim, is presented by Al Borno et al. [16]. OpenSense enables IMU data-based motion analysis using solely the IMU-based motion capture system XSens (Movella Inc., Henderson, NV, USA). OpenSense associates and registers each IMU sensor with a body segment of an OpenSim model. Virtual IMU frames (coordinate systems with three orthogonal axes) are placed onto the musculoskeletal model. OpenSense uses an inverse kinematics approach which—in contrast to the marker-based inverse kinematics approach—minimizes the difference between experimentally measured IMU orientations and the orientations of virtual IMU frames placed on the model to compute joint angles.

A method for analysing captured motion data measured by either marker-based or IMU-based systems in real-time is presented by Stanev et al. [17]. The method is an extension of the method presented by Pizzolato et al. [18], in which a marker-based motion analysis in real-time is presented. Through an additional software architecture, the method allows for real-time inverse kinematics and inverse dynamics calculation in OpenSim. Even though the work focuses on the real-time analysis of motion capture data, the authors also show that their method is able to track IMU-based motion data. Analogous to the OpenSense approach, this method uses direct sensor orientation tracking. It associates and registers each IMU sensor with a body segment. Virtual IMUs are placed onto the musculoskeletal model. Then, the sensor orientations are used in a least-squares approach to track captured motion data. The authors used a custom-built IMU sensor system. They reported technical challenges; errors accumulated in their data because of bias and sensor drift.

Karatsidis et al. [19] exported motion data, captured using the IMU-based system Xsens (Movella Inc., Henderson, NV, USA), to the file format BVH.

A BVH file consists of two parts—a header and a data section. The header section provides information about the hierarchy and the initial pose of the skeleton as well as information about degrees of freedom and rotation sequences. The hierarchy consists of multiple segments that are structured in a parent-child relationship. The initial segment, which has no parent segment, is referred to as the root. In most cases, the pelvis depicts the root. The segments are connected through joints, each consisting of a channel and an offset. The channel defines the number of degrees of freedom of the joint. The offset defines the segment length to this joint and, therefore, the parent segment length. Each joint is named after its child segment. Final joints do not have a following segment and thus are called “EndSite”. Figure 1 exemplarily shows the segment structure of a BVH file. The exact hierarchy structure of a BVH file depends on the IMU-based motion capture system used for motion recording. The following data section describes the channel data for each joint and time frame. In addition, it lists the number of frames and the sampling interval. The data for all defined joints are stored using Euler rotations.

Karatsidis et al. [19] used the exported BVH file to generate a stick figure model, which represents the skeletal system part of the BVH file. Virtual markers are placed in every joint rotation center and outward of each body segment. The stick figure model then executes the previously experimentally measured motion, and the position of each virtual marker at every time step is saved in a separate file. A set of markers corresponding to those of the stick figures is placed on a musculoskeletal model constructed in the Anybody Modeling System (Anybody Technology, Aalborg, Denmark). The virtual markers of the stick figure model are then treated the same way as experimental markers from optical motion capture systems. During the motion capture process, a static pose has to be captured in order to scale the musculoskeletal model. Using the marker file depicting the static pose, a generic musculoskeletal model is scaled. Afterwards, an inverse kinematics analysis is conducted. For each time step, the distance between virtual and model markers is minimized by a least squares method.

### 1.3. Research Gap & Objective

Currently, only tools for specific IMU full-body measurement systems are available in order to transfer IMU data to musculoskeletal models in OpenSim. Other solution approaches enable the transfer of IMU-based motion data into musculoskeletal simulation programs other than OpenSim (e.g., the Anybody Modeling System). Up to now, there is no universal way to transfer IMU data from arbitrary full-body IMU measurement systems into arbitrary digital human modelling software. The objective of this study was to enable the transfer of collected IMU motion data stored as a BVH file to the digital human modelling software OpenSim 4.4 to visualize and analyse the motion using musculoskeletal models. We verified our method for motion using both upper and lower body motions. For the upper body, we analysed an arm-lifting and a reaching motion. For the lower body, we investigated a squat motion. Further, we evaluated the accuracy of our IMU-based motion capture system.

## 2. Materials and Methods

### 2.1. BVH-Based Inverse Kinematics

The method for transferring and analysing motion stored in the BVH file format to a digital human modelling software is described below. The method is presented using the OpenSim 4.4 software as an example. To simplify the description, we have divided the method into four subsections (see Figure 2). A generic musculoskeletal OpenSim model and experimentally measured motion data stored in BVH format serve as initial input for the proposed method.

(1)In the first step, the header part of the BVH file is used to create a stick figure model in OpenSim, which represents the skeleton information of the BVH file. Hereby, each segment of the skeleton is represented by an ellipsoid. The initial joint is the root joint, which is implemented as a 6 degrees of freedom (DOF) free joint between the model and the ground. Each following joint is implemented as a 3 degrees of freedom ball joint. Thus, the exact number of degrees of freedom of the model depends on the number of joints, which again depends on the skeleton hierarchy of the BVH file. After the stick figure model has been created, virtual markers are placed onto the stick figure model. The markers are placed into the rotation centers of each joint, and for each segment, one virtual marker is placed outward of the segment in order to be able to measure translations and rotations for every coordinate axis (see Figure 3).(2)The motion data contained in the BVH file is converted into the sto file format, which is readable by OpenSim. For that, the data are recalculated to match the OpenSim ball joint definition. For each 3 DOF joint, the sto file contains three joint angle values for every time frame. For the root joint, information about 6 degrees of freedom is stored (three joint angle values and three translation values).(3)Using the sto motion data, the stick figure can execute the experimentally measured motion. For each time step, the position of each virtual marker with respect to the global coordinate system is extracted and saved into a trc marker file. This file then corresponds to marker trajectory files measured by a conventional marker-based motion capture process.(4)To scale the generic musculoskeletal model, motion data measurements of a person standing in static T-pose are necessary. These data can then be used to perform the conventional marker-based scaling approach of OpenSim. In order to do that, markers corresponding to the virtual markers placed onto the stick figure have been placed on a generic musculoskeletal OpenSim model. The joint markers of the stick figure are placed in the origin of body frames of the model in which the corresponding joints are defined. The stick figure’s segment markers are placed in body frames so that their position is perpendicular to the connecting line of the joint markers between which the segment marker is placed (see Figure 3). The virtual markers are then used analogously to experimental marker data. Consequently, the generic musculoskeletal model is scaled by the marker data extracted from the OpenSim stick figure in the previous step. Each segment of the musculoskeletal model is scaled such that the distance between model markers ($m_{i}$) matches the distance between the virtual markers ($e_{i}$) on the OpenSim stick figure model. To do so, scaling factors ($s_{i}$) are computed using Equation (1) (1)$$s_{i}=\frac{e_{i}}{m_{i}}$$ Afterwards, an inverse kinematics analysis is conducted.

### 2.2. Participants

The experimental data were collected at the motion capture laboratory of the Institute of Engineering Design of Friedrich-Alexander-Universität Erlangen-Nürnberg. Three participants (age: 26.6 ± 1.1 years; height: 1.698 ± 0.825 m; weight: 64.7 ± 9.0 kg) volunteered for the study. Written consent was provided by all participants prior to data collection.

### 2.3. Instrumentation

The IMU-based motion capture system Perception Neuron Studio (Noitom Ltd., Beijing, China) was used in this study. A total of 17 sensors were placed on the head, the upper arms, the forearms, the hands, the upper back, the pelvis, the thighs, the shanks and the feet of each participant using size-adjustable straps. The placement of the sensors followed the guidelines given in the user manual of the system [20]. Data recording was done by using the corresponding software Axis Studio [21] of the Perception Neuron system. The data were recorded with a sampling rate of 100 Hz and exported as a BVH file.

### 2.4. Experimental Protocol

Manual measurements of the participants’ anthropometry were taken before the motion capture process took place. These data were used as input in the Axis Studio software in order to scale the generic mannequin model of Axis Studio for each participant. Measured dimensions included: palm length, forearm length, upper arm length, shoulder width, hip width, head length, neck length, torso length, upper and lower leg length, ankle height and foot length. Additionally, the following functional body dimensions were measured: body height, inseam height, arm span width and grip height while standing.

Each participant completed the following motion tasks: squat, raising arms into T-pose and a reaching motion. For the reaching motion, the subjects stood in front of a box with their feet 0.21 m apart. The distance between the heel of the feet to the edge of the box amounted to 0.68 m. The dimensions of the box were 0.35 × 0.33 × 1.34 m (length, width, height). Markers on the floor indicated the desired foot position. The arms were lifted upwards until the middle finger of the hand was at the same height as the top edge of the box. The measurement setup is shown in Figure 4. The sensor orientations of each IMU of the Perception Neuron system were calibrated for each participant before the motion-capturing process using the A-pose, T-pose and S-pose. Each motion was recorded separately. While the measurements took place, the participants started each motion from the A-pose. Each motion type was recorded twice—once with a single repetition and again with five repetitions.

### 2.5. Perception Neuron BVH Model

The BVH model, which is exported by the Perception Neuron System, has 60 joints in total. The system is able to measure detailed finger motion in the case where special motion-capturing gloves are used. As exact finger motions were not of interest for this work, we did not use the measurement gloves. Thus, our model has 22 moveable joints.

### 2.6. Verification Measures

We verified our method using both qualitative and quantitative measures. To detect possible error sources and weaknesses of the method, each step of the method was evaluated separately. First, the manual measurements, the BVH file and the OpenSim stick figure were compared quantitatively using specific segment lengths measured both in the BVH file and the OpenSim stick figure model. Investigated segment lengths included: body height, upper and lower leg length, foot (ankle) height, torso length, upper arm length, forearm length, palm length and head length. The segment lengths of the BVH stick figure can be taken directly from the BVH file itself. The segment lengths of the OpenSim stick figure model were computed by obtaining the joint marker positions in the OpenSim ground frame. We performed this comparison to verify that the Axis Studio Software does work as intended and no information is distorted or lost through the method. Additionally, the manual measurements, the BVH file and the OpenSim musculoskeletal model were compared using the same specific segment lengths. This was carried out to verify that the method is able to scale the generic musculoskeletal model correctly according to the body dimension values saved in the BVH file. Next, the position of the stick figure’s virtual markers and their corresponding positions in the generated trc marker file were qualitatively examined. For that, the stick figure model was loaded into the OpenSim graphical user interface, and the motion file—and simultaneously the trc marker file—was loaded into the model. Before the motion was investigated quantitatively, a first visual inspection of the corresponding marker positions examined the agreement with the model for the whole motion. In order to confirm that the motion stored in the BVH file was correctly transferred to the OpenSim stick figure model, the stick figure’s motion in OpenSim and the original BVH motion were compared qualitatively by visually comparing the pose of the models at certain points during the motion. Both motions were also quantitatively compared by comparing specific distances at certain points during the motion. The absolute distances between wrist joints, knee joints and ankle joints were compared for one participant for two poses during a squat motion. We also analysed the overall anthropometric error, which describes the difference between manually measured body dimension values of the participants and the corresponding values of the musculoskeletal model. Defined body lengths were chosen to compare the model with reality to quantify the size of the anthropometric error. For that, we compared functional dimensions, values that were not used for the scaling approach in the method, including the overall body height, the inseam height, the grip height while standing and the arm span. To evaluate the quality of the kinematic data transfer from the stick figure to the musculoskeletal model, the root mean square error (RMSE) and the maximum error of the inverse kinematics method were investigated. Additionally, position coordinate values of a marker on the tip of the middle finger (left and right hand) stored in the trc marker file and the position of the corresponding markers in the motion file, generated by the inverse kinematics approach, were compared. To evaluate the accuracy of the IMU-based motion capture system, the location of a hand grip marker within the musculoskeletal model was compared with the known position of the hand in reality. The position of the hand in reality was measured for each participant before the motion capture process took place. On the musculoskeletal model, the hand grip marker lies on the intersection of the grip axis of the palm and the longitudinal axis of the middle finger. Joint angle trajectories of both the lower and upper body were also investigated. Lower limb joint angle trajectories (hip, knee and ankle flexion angle) were analysed for the squat motion and joint angle trajectories of the upper limb joints (shoulder, elbow, wrist) were analysed for the reaching and the arm-raising motions to evaluate whether or not feasible motion results were generated.

## 3. Results

### 3.1. Anthropometric Measurements

#### 3.1.1. Comparison with Manual Measurements—BVH File

A comparison of specific body dimensions between manual measurements, the skeletal system information of the BVH file and the dimensions of the OpenSim stick figure model showed good agreement between all three values for all participants (see Table 1). The OpenSim stick figure is shown in Figure 3a on the right side. There are no deviations between all three data sources for all body dimensions but one. The palm length was correctly transferred from the BVH file to the stick figure model, but this value differed slightly from the manual measurements.

#### 3.1.2. Comparison with Manual Measurements—Musculoskeletal Model

A comparison of specific body dimensions between manual measurements, the skeletal system information of the BVH file and the dimensions of the musculoskeletal model showed very good agreement between all three values for all participants (see Table 2). The OpenSim musculoskeletal model is shown in Figure 3a on the left side. For all participants, comparison between functional dimensions of the musculoskeletal model and manual measurements showed very good agreement for all investigated dimensions (body height, inseam height, arm span width and grip height). All deviations were smaller than 1 cm (see Table 3).

### 3.2. Motion Extraction & Transfer

A qualitative comparison of the BVH file and the OpenSim stick figure model at two certain points in time during the squat motion showed good agreement. For both points in time, the poses of the OpenSim stick figure model and the BVH file match very well (see Figure 5). A qualitative comparison of virtual model marker positions and trc file marker positions showed optimal agreement; see the OpenSim stick figure models in Figure 5. Corresponding virtual model markers (shown in pink) and markers from the trc file (shown in blue) were in the same position. The quantitative comparison between wrist joint marker, knee joint marker and ankle joint marker distances on the BVH file and OpenSim stick figure model for one participant at two different points during the motion showed good agreement. For both investigated poses, the difference between distances of all points were negligible (smaller than 0.5 cm in all cases) (see Table 4).

### 3.3. Hand Position Analysis

Table 5 shows the comparison of position coordinate values of the middle finger markers (left and right) between the positions stored in the trc file and the position of the corresponding markers in the motion file, generated by the inverse kinematics method. Values depict the mean of the position of the middle finger markers at the five upmost positions during the lifting motion with five repetitions. All values are expressed in the global coordinate system. The values showed very good agreement for all coordinates, as all deviations were smaller than 1 cm.

Table 6 shows the comparison of hand position coordinate values between the OpenSim model and reality. Model values are the mean of the upmost position of the hand during the reaching motion with five repetitions. Values for grip height during the reaching motion overall showed reasonable agreement for participant 3 (maximum deviation of 8 cm). For participant 2, values showed reasonable agreement for the Y and Z coordinates (maximum deviation of 8 cm). The X coordinate differed about 11 cm for the right hand and 9 cm for the left hand. For participant 1, values showed reasonable agreement for both the X and Y coordinates of both hands (maximum deviation of 7 cm). For the Z coordinate of the right hand, deviations were highest overall (deviation of 19 cm).

### 3.4. Kinematic Transferability

After verifying that our method does correctly transfer both the skeletal and the motion data stored in the BVH file onto the OpenSim stick figure model, the results of the inverse kinematics method were evaluated. For the squat, the reaching and the arm-lifting motion, the RMSE was smaller than 0.0126 m, 0.0112 m and 0.0104 m, respectively. Maximum errors were smaller than 0.0329 m, 0.0292 m and 0.0246 m, respectively. The trajectories of the most relevant joints and degrees of freedom for each of the three investigated motions (squat, reaching and arm-raising) are presented. All joint angle trajectories have been filtered using a third-order Butterworth filter with a cut-off frequency of 6 Hz. Figure 6 shows the hip, knee and ankle flexion angle trajectories for one participant during the squat motion with five repetitions. Maximum and minimum angles stayed consistent over each repetition.

Figure 7 shows shoulder angles for one participant during the reaching motion with five repetitions. Our musculoskeletal model includes the dynamic upper limb model presented by Saul et al. [22]. The position of the arm is determined by the rotation angle, the elevation angle and the angle of the plane of elevation. The plane of elevation does not describe a physiological shoulder angle, but the position of the elevation plane relative to the frontal plane (see Figure 8). For the shoulder elevation angle, maximum and minimum angles stayed consistent. The amplitudes of the plane of shoulder elevation and shoulder rotation angles varied over the five repetitions.

Figure 9 shows upper body joint angles for one participant during the arm-raising motion with five repetitions. Shoulder elevation maximum and minimum joint angle values stayed consistent over all repetitions. Elbow flexion angle ranged between 5° flexion and 10° extension. The wrist flexion angle remained at 0° throughout the motion.

## 4. Discussion

In this paper, we presented an easily applicable method that enables the transfer of BVH files measured with an IMU-based motion capture system to musculoskeletal models in OpenSim 4.4. The method (shown in Figure 2) was verified using different qualitative and quantitative verification measures. Specific segment lengths between the BVH model and the OpenSim stick figure model and the musculoskeletal model were compared to ensure that our method generates a correctly sized stick figure model (Step 1) (see Table 1 and Table 2). All values showed very good agreement. Afterwards, virtual markers were placed onto the stick figure model. The motion part of the BVH file was extracted, and the given angles were converted in order to fit to joint angle definitions in OpenSim. The data were then exported as a sto file, which stores the experimental motion data in generalized coordinates (joint angles) (Step 2). Using the extracted and stored motion, the OpenSim stick figure executed the motion, and for every time step, the global positions of each virtual marker were extracted and stored in a trc marker file (Step 3). Then, the marker positions in the trc file were compared with corresponding initial virtual marker positions (Step 3). The comparison of specific distances (wrist, knee and ankle joint distances) both in the BVH file and the OpenSim stick figure model during one motion showed good agreement (see Figure 4 and Table 4). The captured motion was accurately transferred. A comparison between functional body dimensions, measured using a measuring tape, and the corresponding measurements of the musculoskeletal models also showed good conformity (Step 4) (see Table 3). For all investigated motions, RMSEs were small (≤0.0126 m). A comparison of the position coordinate values of middle finger markers between the positions stored in the trc file and the motion file showed negligible deviations (≤0.01 m) (see Table 5). This indicates that the kinematic data are correctly transferred from the BVH file to the musculoskeletal OpenSim model. To evaluate the accuracy of our IMU-based motion capture system, a reaching motion was captured, and the hand positions between the model and reality were compared (see Table 6). Deviations between both values were mostly small. For both upper and lower limb motions, joint angle trajectories were overall consistent (see Figure 6, Figure 7 and Figure 9). Overall, the method enables the transfer of motion capture data, captured using arbitrary IMU systems, to musculoskeletal models, provided that the IMU system is able to export the data in BVH format. The results show that the method works as intended, and the measured motion data are correctly transferred onto the musculoskeletal model in OpenSim 4.4 without resulting in data loss.

The main drawback of our approach is its reliance on the BVH file format. The simplified joint and segment definition of the BVH file results in data loss and an unrealistic representation of the human locomotor apparatus. The joint definitions differ slightly between the BVH file (or stick figure) and the musculoskeletal model. The BVH model consists of only three DOF ball joints. Naturally, this does not depict the physiologically realistic human shoulder joint. The musculoskeletal model attempts to recreate the shoulder joint using joint definitions that are as physiologically realistic as possible. Our musculoskeletal model includes the dynamic upper limb model presented by Saul et al. [22]. Hereby, the shoulder joint is implemented as a coupled joint, as the range of motion of the shoulder joint is determined by the movement of the shoulder girdle (humerus, scapula and clavicle) [23]. Because of its joint definition, the BVH model is able to reach or generate specific joint angles or poses that cannot be depicted by the physiological human body and, thus, the musculoskeletal model. As a consequence, the musculoskeletal model is not able to track the virtual marker trajectories without errors; deviations can be observed. However, the deviations are overall quite small and deviations between the marker positions of the end effectors (the hands) are nearly negligible (see Table 5). Nevertheless, for future work, a reduction in the size of the RMSE and the maximum error should be attempted. As the deviations stem from the difference between the stick figure and the musculoskeletal model, an enhancement of the stick figure model, in the sense of aligning the stick figure model with the musculoskeletal model, could lead to better marker tracking performance.

When looking at the joint angle trajectories resulting from the inverse kinematics method, the following is noticeable. For the arm-raising motion, the shoulder elevation angle shows a smooth cyclic trajectory with consistent minimum and maximum angle values (see Figure 9). As expected, the wrist flexion angle was null over the whole motion duration. The elbow flexion angle showed a varying trajectory that ranged between five degrees flexion and ten degrees extension. For the reaching motion, the shoulder elevation angle showed a cyclic trajectory with consistent minimum and maximum joint angles (see Figure 7). The plane of elevation angle showed a cyclic trajectory, but with varying maximum and minimum values. Additionally, the shoulder rotation angle did not show a cyclic trajectory. Both the reaching and the arm-lifting motion were executed with fully extended arms. Because of this, the model cannot correctly resolve what proportion of rotation of the arm is generated by the upper arm or the forearm. Varying shoulder rotation and plane of elevation angles are the result.

The accuracy of our IMU-based motion capture system in combination with our method was evaluated by comparing hand positions between the musculoskeletal model and reality for a reaching motion. As most deviations between model and reality were of reasonable size, one can say that our method generates decent results. But, for participant 1, deviations for the right hand were quite large (19 cm). When comparing the video material—which was taken during the motion capture process—with the captured IMU data, one can see that the participant followed the given instructions directly, because his right hand clearly touches the upper edge of the box (see Figure 10). Nevertheless, the comparison between global positions of middle finger markers between the trc marker file and the motion file showed minimal deviations for all coordinates. This shows that our method exactly and correctly transfers the motion data that are stored in the BVH file onto the OpenSim model, but it also shows that our method is susceptible to problems that are inherent to IMU-based motion capture systems. The quality of the results of IMU-based motion capture systems depends on the performance of the calibration procedure. However, IMU systems are fundamentally subject to calibration errors. If the calibration fails, was not executed well enough or was influenced by environmental disturbances (e.g., magnetic fields), the quality of the captured motion data will be negatively influenced. The deviation between reality and the captured motion probably stems from a calibration error. One potential error source could be an offset between the desired and actually executed calibration pose. If the executed pose does not match the desired pose well enough, the system may be calibrated with this offset. Even though we took care to calibrate our system as well as possible, we still received erroneous measurement data. In addition to the calibration problem, IMU measurements are, in general, susceptible to sensor noise and drift. This is because IMU-based measurement systems, in comparison to marker-based systems, do not include an absolute (or global) reference. Therefore, a possible solution to compensate for measurement errors could be using multimodal data. Extending the present method to include multimodal motion capture data could lead to more accurate motion capture results. A second or third type of motion measurement could compensate for the drift and inaccurate position values and thus make IMU-based motion measurements more accurate and reliable.

Our experimental protocol had two main limitations—a small sample size and a limited number of analysed motions. Additionally, it was not possible to use a marker-based motion capturing system as a reference system. Because of this, another experimental study is needed to validate our approach and to obtain more detailed knowledge about the joint angle result accuracy of IMU-based motion capture systems in combination with the present method.

## 5. Conclusions

We presented an easily applicable method that enables the transfer of motion data captured with an IMU-based measurement system and stored in BVH file format to musculoskeletal models in OpenSim 4.4. We extracted the skeletal system information that is stored in the BVH file to generate a corresponding stick figure model in OpenSim. Virtual markers were placed onto the stick figure model. Using the motion data stored in the BVH file, we generated a virtual marker file. Afterwards, this marker file was used analogous to a conventional experimental marker file. First, a generic musculoskeletal model was scaled. After that, an inverse kinematics analysis was conducted. The method generated satisfactorily good results, even though our approach was influenced by some limitations. The body dimensions of the resulting musculoskeletal models corresponded very well to the skeletal information of the BVH file. The captured motion was correctly and reliably transferred from the BVH file to the musculoskeletal model. Joint angle trajectories were overall consistent, and marker errors were overall small.

In the next steps, we want to further enhance the method and reduce limitations. We want to conduct another experimental study with a larger sample size and simultaneously capture both IMU- and marker-based motion capture data to validate the kinematic and dynamic results of the method. Additionally, we want to investigate if the extension of the method to multimodal motion measurements in order to compensate for errors such as measurement noise and sensor drift leads to more reliable and accurate motion capture results. The effect of the alignment of the stick figure model to the musculoskeletal model on the quality of the kinematic results is also to be investigated. We now have a concrete approach available that enables us to transfer IMU-based motion capture data onto a musculoskeletal model in OpenSim. IMU-based motion capture systems are less expensive and generally more easily applicable than marker-based systems. As our approach uses the BVH file format to transfer the data, it is independent from the motion capture measurement system used. As long as the system is able to export the motion data as a BVH file, our approach can be used to analyse IMU data using a musculoskeletal model. Thus, this makes musculoskeletal models more accessible for people who do not have access to a marker-based motion capture system or whose activities of interest cannot be measured in a gait laboratory.

## Figures and Tables

**Figure 1 sensors-23-05423-f001:**
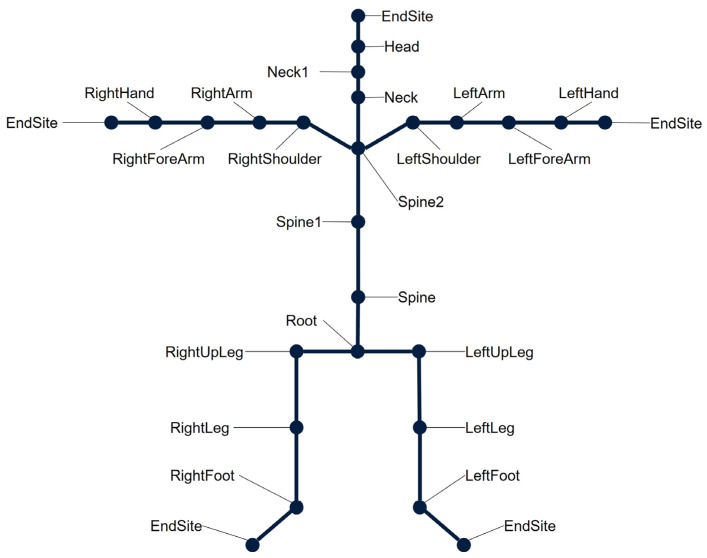
Exemplary hierarchical skeletal structure of a BVH file. The exact structure depends on the IMU-based motion capture system. The skeletal structure consists of segments that are connected by 3 DOF ball joints. Each joint is named after its child segment. As final joints do not have a child segment, they are called Endsite.

**Figure 2 sensors-23-05423-f002:**
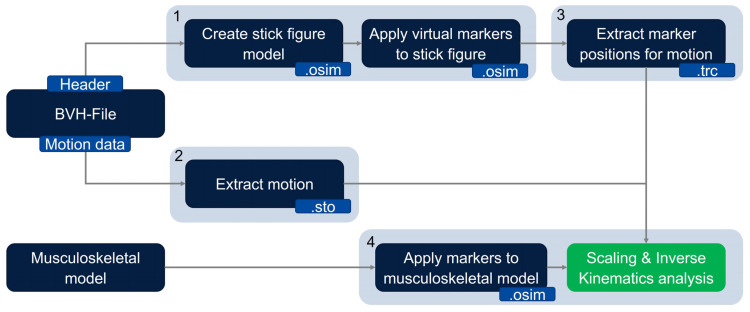
Method for transferring and analysing motion, captured using IMU-based systems and stored in BVH file format in OpenSim 4.4.

**Figure 3 sensors-23-05423-f003:**
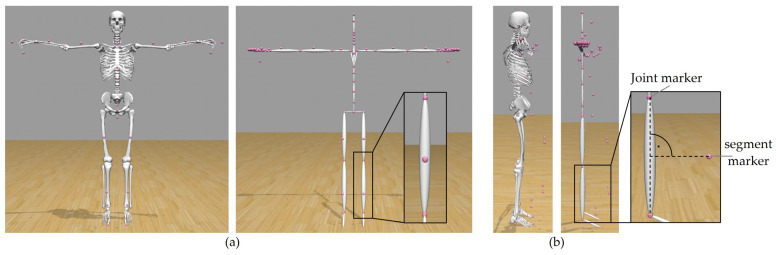
Placement of virtual model markers (pink) on stick figure and musculoskeletal model in (**a**) frontal and (**b**) sagittal plane.

**Figure 4 sensors-23-05423-f004:**
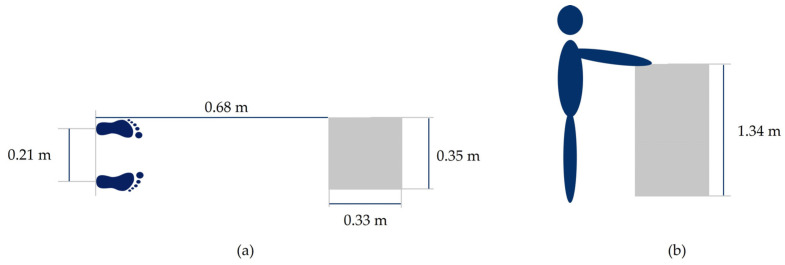
Measurement setup: (**a**) top view and (**b**) side view.

**Figure 5 sensors-23-05423-f005:**
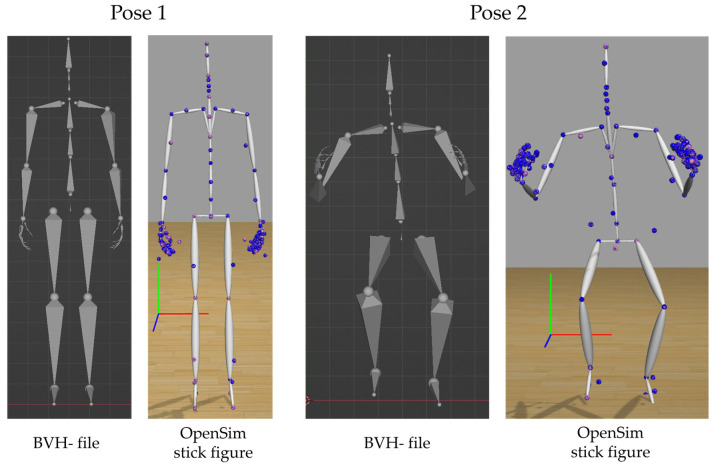
Comparison between the BVH file, imported in the software Blender, and the stick figure model in OpenSim 4.4. On the OpenSim model, virtual model markers are shown in pink. Markers saved in the trc file are shown in blue. Both in pose 1 and pose 2 all corresponding markers lie in the same position. Specific distance (wrist, knee and ankle joint distances) values differ only slightly between the BVH file and the OpenSim stick figure model. The global coordinate system in OpenSim (red: x-, green: y- and blue: z-direction) is shown.

**Figure 6 sensors-23-05423-f006:**
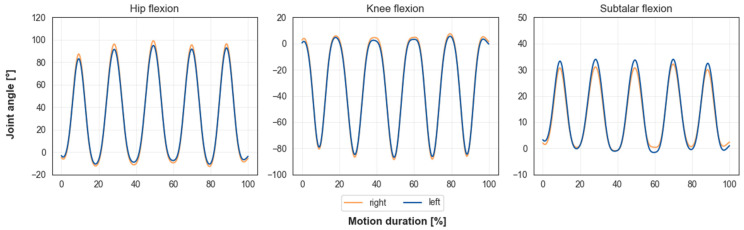
Lower body joint angles of one participant for the squat motion with five repetitions. Trajectories are smooth and in physiological ranges. Maximum and minimum angle values for each joint stay consistent over all repetitions.

**Figure 7 sensors-23-05423-f007:**
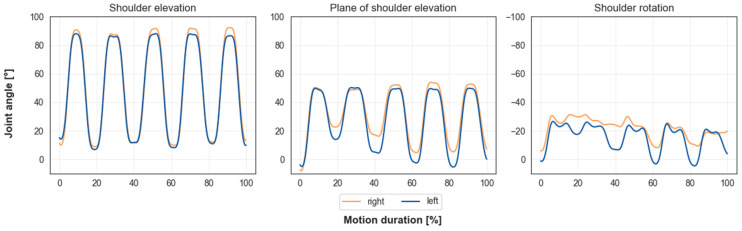
Shoulder joint angles for the reaching motion with five repetitions for one participant. Trajectories are smooth and cyclic. Maximum and minimum angle values stay consistent for shoulder elevation angle but vary for the plane of shoulder elevation angle and shoulder rotation angle.

**Figure 8 sensors-23-05423-f008:**
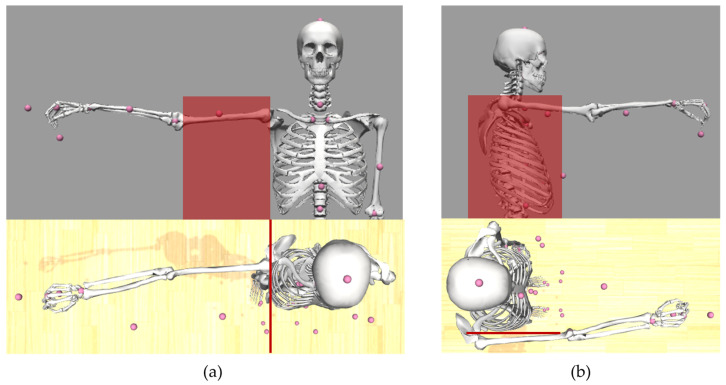
Arm position for 90° elevation angle and a plane of elevation angle of 0° (**a**) and 90° (**b**). The plane of elevation is shown in red for both cases. The pink dots inidcate the virtual model markers.

**Figure 9 sensors-23-05423-f009:**
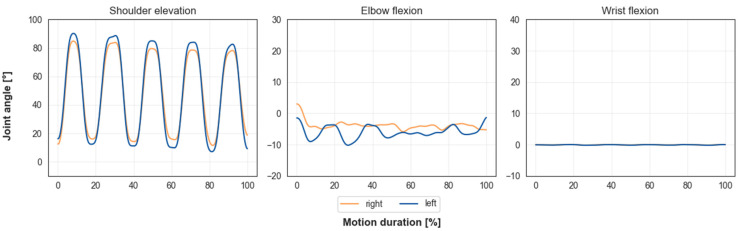
Upper body joint angles of one participant for the arm-lifting motion with five repetitions. Trajectories are smooth for shoulder elevation angle, and maximum and minimum angle values stay consistent over each repetition. Elbow flexion trajectory ranges between 5 degrees flexion and 10 degrees extension. Wrist flexion is null over whole motion duration.

**Figure 10 sensors-23-05423-f010:**
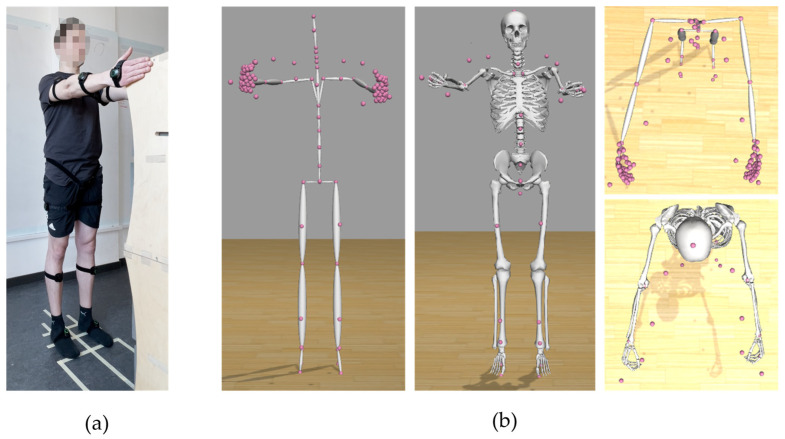
Comparison between body pose of participant 1 during the reaching motion in (**a**) reality and (**b**) the poses of the stick figure and musculoskeletal model. The pink dots indicate the virtual model markers.

**Table 1 sensors-23-05423-t001:** Comparison of specific body dimensions [m] between manual measurements, the BVH model file and the OpenSim stick figure. For all participants, there is no deviation for all body dimensions but one. The palm length values are equal for the BVH file and the OpenSim stick figure model but differ from the manual measurement. The palm length values are highlighted by a grey background.

Body Dimension [m]	Participant 1			Participant 2			Participant 3		
	Manual *	BVH	Stick Figure	Manual	BVH	Stick Figure	Manual	BVH	Stick Figure
Body height	1.790	1.790	1.790	1.630	1.630	1.630	1.680	1.680	1.680
Upper leg length	0.420	0.420	0.420	0.390	0.390	0.390	0.390	0.390	0.390
Lower leg length	0.450	0.450	0.450	0.385	0.385	0.385	0.390	0.390	0.390
Ankle height	0.075	0.075	0.075	0.065	0.065	0.065	0.070	0.070	0.070
Torso length	0.590	0.590	0.590	0.540	0.540	0.540	0.580	0.580	0.580
Upper arm length	0.280	0.280	0.280	0.260	0.260	0.260	0.280	0.280	0.280
Forearm length	0.260	0.260	0.260	0.240	0.240	0.240	0.250	0.250	0.250
Palm length	0.190	0.184	0.184	0.170	0.165	0.165	0.180	0.174	0.174
Head length	0.155	0.155	0.155	0.150	0.150	0.150	0.145	0.145	0.145

* Manual: manual measurements.

**Table 2 sensors-23-05423-t002:** Comparison of body dimensions between manual measurements, BVH file and the musculoskeletal model. For all participants, there is no deviation for all body dimensions but one. The palm length values are equal for the BVH file and the musculoskeletal model but differ from the manual measurement. The palm length values are highlighted by a grey background.

Body Dimension [m]	Participant 1			Participant 2			Participant 3		
	Manual	BVH	OpenSim *	Manual	BVH	OpenSim	Manual	BVH	OpenSim
Body height	1.790	1.790	1.790	1.630	1.630	1.630	1.680	1.680	1.680
Upper leg length	0.420	0.420	0.420	0.390	0.390	0.390	0.390	0.390	0.390
Lower leg length	0.450	0.450	0.450	0.385	0.385	0.385	0.390	0.390	0.390
Ankle height	0.075	0.075	0.075	0.065	0.065	0.065	0.070	0.070	0.070
Torso length	0.590	0.590	0.590	0.540	0.540	0.540	0.580	0.580	0.580
Upper arm length	0.280	0.280	0.280	0.260	0.260	0.260	0.280	0.280	0.280
Forearm length	0.260	0.260	0.260	0.240	0.240	0.240	0.250	0.250	0.250
Palm length	0.190	0.184	0.184	0.170	0.165	0.165	0.180	0.175	0.175
Head length	0.155	0.155	0.155	0.150	0.150	0.150	0.145	0.145	0.145

* OpenSim: musculoskeletal model.

**Table 3 sensors-23-05423-t003:** Comparison between musculoskeletal model dimensions and manual measurements. All body dimensions show very good conformity.

Dimension [m]	Participant 1		Participant 2		Participant 3	
	Manual	Model	Manual	Model	Manual	Model
Body height	1.790	1.793	1.630	1.640	1.680	1.679
Inseam height	0.870	0.880	0.770	0.781	0.780	0.787
Grip height (r)	0.820	0.817	0.750	0.760	0.740	0.742
Arm span width	1.770	1.779	1.600	1.596	1.685	1.694

**Table 4 sensors-23-05423-t004:** Comparison of specific distances [m] between the BVH model file and the OpenSim stick figure at two different points in time during the squat motion. Values show good agreement for both poses.

Pose	Wrist Joint Marker		Knee Joint Marker		Ankle Joint Marker	
	Stick Figure	BVH	Stick Figure	BVH	Stick Figure	BVH
1	0.480	0.480	0.163	0.163	0.172	0.172
2	0.612	0.607	0.314	0.310	0.248	0.244

**Table 5 sensors-23-05423-t005:** Comparison of middle finger marker [m] between the positions stored in the trc file and the motion file and their absolute difference Δ. All deviations are smaller than 1 cm. The absolute difference is highlighted in bold, as it is the measure used for verification.

Participant	Modality	X		Y		Z	
		Left	Right	Left	Right	Left	Right
1	Trc file	0.6811	0.0392	1.2959	1.3508	2.0950	2.1172
	IK result	0.6762	0.0414	1.2925	1.3567	2.0988	2.1186
	**Δ**	**0.0050**	**−0.0021**	**0.0034**	**−0.0059**	**−0.0039**	**−0.0013**
2	Trc file	0.1293	−0.2452	1.3071	1.3632	0.6863	0.6980
	IK result	0.1278	−0.2465	1.3058	1.3635	0.6858	0.6978
	**Δ**	**0.0015**	**0.0013**	**0.0013**	**−0.0003**	**0.0005**	**0.0002**
3	Trc file	0.3640	−0.0474	1.2967	1.3233	0.1912	0.1960
	IK result	0.3627	−0.0411	1.2951	1.3233	0.1924	0.1963
	**Δ**	**0.0013**	**−0.0063**	**0.0016**	**0.0000**	**−0.0012**	**−0.0003**

**Table 6 sensors-23-05423-t006:** Comparison of hand position [m] between model and reality and their absolute difference Δ. Model values are the mean of the upmost position of the hand during the reaching motion with five repetitions. All values are expressed in the coordinate system of the right foot (calcaneus). The orientation of the coordinate system of the calcaneus is analogous to the global coordinate system shown in Figure 5. Values show mostly reasonable agreement. The absolute differences is highlighted in bold, as it is the measure used for verification.

Participant	Modality	X		Y		Z	
		Left	Right	Left	Right	Left	Right
1	Model	0.69	0.69	1.32	1.37	−0.33	0.27
	Actual	0.62	0.62	1.34	1.34	−0.28	0.08
	**Δ**	**0.07**	**0.07**	**0.02**	**0.03**	**−0.05**	**0.19**
2	Model	0.69	0.67	1.26	1.31	−0.25	0.11
	Actual	0.58	0.58	1.34	1.34	−0.28	0.08
	**Δ**	**0.11**	**0.09**	**−0.08**	**−0.03**	**0.03**	**0.03**
3	Model	0.70	0.61	1.28	1.31	−0.30	0.09
	Actual	0.62	0.62	1.34	1.34	−0.28	0.08
	**Δ**	**−0.08**	**0.01**	**−0.06**	**−0.03**	**−0.02**	**0.01**

## Data Availability

The data presented in this study are available on request from the corresponding author. The data are not publicly available due to privacy restrictions.

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
