# Peer review of "Method for Using IMU-Based Experimental Motion Data in BVH Format for Musculoskeletal Simulations via OpenSim"

_sensors, 2023, doi:10.3390/s23125423_

Round 1
Reviewer 1 Report
This was a well written technical note describing the process of using IMU data to run OpenSim simulations. I think this will be of interest to the musculoskeletal modeling community. A few minor edits would strengthen this manuscript and prepare it for publication.
Lines 54-55: This sentence makes it sound like IMU use is more popular than traditional marker based motion capture. Consider rephrasing. Suggest: During the last decade, the popularity of wearable IMUs has increased.
Line 71, 81, 82, 92, 108, 370, 433: The last name of the Author, et. al. should be included in the sentence in addition to just the reference number.
It took me a while to figure out the difference between Table 1 and Table 2. Clarity on Stick Figure measurements vs OpenSim measurements is needed.
Lines 341-342: The left hand is listed twice. Assuming one measure should be for the right hand.
Reviewer 2 Report
This paper proposed a framework to transfer the collected IMU motion data stored as BVH file to OpenSim for the visualization and analysis of the motion using musculoskeletal models. The paper was well written. The related works as well as the proposed scheme are presented in detail. The performance indicates that the captured motions could be correctly and reliably transferred from the BVH file to the models. A minor problem is that as the authors claimed there is no universal way to transfer IMU-data from arbitrary full-body IMU measurement systems into arbitrary digital human modelling software, is there any way to do a quantitative analysis among your IMU-based system to other state-of-the-art systems based on marker-based optical ones?
